# Biohybrid lung Development: Towards Complete Endothelialization of an Assembled Extracorporeal Membrane Oxygenator

**DOI:** 10.3390/bioengineering10010072

**Published:** 2023-01-05

**Authors:** Hussam Almesto Alabdullh, Michael Pflaum, Marisa Mälzer, Marcel Kipp, Hossein Naghilouy-Hidaji, Denise Adam, Christian Kühn, Russlan Natanov, Adelheid Niehaus, Axel Haverich, Bettina Wiegmann

**Affiliations:** 1Department for Cardiothoracic, Transplantation and Vascular Surgery, Hannover Medical School, 30625 Hannover, Germany; 2Lower Saxony Center for Biomedical Engineering, Implant Research and Development (NIFE), Stadtfelddamm 34, 30625 Hannover, Germany; 3German Center for Lung Research (DZL), Carl-Neuberg-Str. 1, 30625 Hannover, Germany

**Keywords:** biohybrid lung, ECMO, oxygenator, endothelialization, tissue engineering

## Abstract

Towards the establishment of a long-term lung-assist device to be used both as a bridge and as an alternative to lung transplantation according to final destination therapy, we develop the biohybrid lung (BHL) on the technical basis of contemporary extracorporeal membrane oxygenation (ECMO). Here, to overcome the significant drawbacks of ECMO, in particular the missing hemocompatibility of the artificial surfaces, all blood-contacting areas need to be endothelialized sufficiently. In continuation of our recent accomplishments, demonstrating the feasibility of establishing a physiological acting endothelial cell (EC) monolayer on the hollow fiber membranes (HFMs) of the ECMO in vitro, the next step towards BHL translation is the endothelialization of the complete oxygenator, consisting of HFMs and the surrounding housing. Therefore, we assessed EC seeding inside our model oxygenator (MOx), which simulated the conditions in the assembled HFM oxygenators in order to identify the most important factors influencing efficient endothelialization, such as cell seeding density, cell distribution, incubation time and culture medium consumption. Overall, upon adjusting the concentration of infused ECs to 15.2 × 10^4^/cm^2^ and ensuring optimal dispersion of cells in the MOx, viable and confluent EC monolayers formed on all relevant surfaces within 24 h, even though they comprised different polymers, i.e., the fibronectin-coated HFMs and the polysulfone MOx housing. Periodic medium change ensured monolayer survival and negligible apoptosis rates comparable to the reference within the assembled system. By means of these results, revealing essential implications for BHL development, their clinical translation is coming one step closer to reality.

## 1. Introduction

Today, lung transplantation (LTx) is the only curative therapy option for patients diagnosed with an end-stage lung disease (ELD) [1]. However, the number of available donor lungs meeting the requirements for LTx [2] significantly falls below the number of patients in need [2], and, thus, LTx is only available for rigorously selected patients [3,4]. Due to annually increasing prevalence and incidence of ELD, also ranking ever higher in the top ten causes of death worldwide [5], the ever-increasing necessity for the development of novel and innovative therapeutic approaches steadily grows.

Generally, in the event of respiratory failure and insufficient artificial ventilation, extracorporeal membrane oxygenation (ECMO) can be applied as a bridge to regeneration or transplantation [6]. However, despite significant developments, for example, anti-coagulative surface coatings (e.g., heparin/albumin) [7] and systemic anti-coagulative therapy, ECMO can only be considered as a temporary interim solution for several days or weeks rather than as a long-term or final destination therapy. This is due to the inevitable contact of the circulating blood with the artificial surfaces (e.g., gas exchange membranes or oxygenator housing), which causes severe complications by adsorption of blood proteins and coagulation factors [8,9], inducing thrombus formation, inflammation and activation of the complement system [10], which results in near-term device occlusion and ECMO failure.

Aiming towards a life-saving therapy alternative to LTx, according to the final destination therapy, such as left ventricular assist devices [11] for end-stage heart diseases, we focus on the development of the biohybrid lung (BHL). In this process, we utilize the efficient gas exchange technology of ECMO combined with tissue engineering methods to secure complete device hemocompatibility for sufficient long-term application. Furthermore, we intend to cover all blood-contacting surfaces with a viable, confluent and self-regenerating endothelial cell monolayer (EML), which not only masks the pro-thrombogenic artificial surfaces, but also actively prevents hemostasis by the expression of anti-thrombogenic surface molecules [12] in a manner comparable to their perceived function in the native blood vessels, rendering systemic anti-coagulative therapy unnecessary. Encouraging results using this approach have been already described for other blood-contacting devices, e.g., vascular prostheses [13,14]. Thus, BHL also has the potential to replace current ECMO therapy as a safer bridge to recovery or transplantation.

Towards BHL development, in recent years, promising research has provided evidence of its overall feasibility, in particular, presenting optimized solutions for specific challenges during the establishment of viable and anti-thrombogenic EML on BHL surfaces. For example, the sufficient endothelialization of the originally hydrophobic and cell-repellent gas exchange hollow fiber membranes (HFMs) made of polymethylpentene (PMP), by identifying optimal seeding protocols and suitable surface coatings, has significantly enhanced endothelial cell (EC) adhesion and the formation of a flow-resistant EML [15,16,17,18,19,20,21,22,23]. Additionally, sufficient EC sources with high relevance for clinical BHL translation were identified, as they facilitate the generation of the required large number of highly proliferative ECs, which are also accepted by any recipient’s immune system without the need for immuno-suppressive therapy [24,25]. Furthermore, our research findings were translated to ECs from animal origin [26] to carry out prospective large animal experiments.

Based on our far-reaching experiences, we now take the next step towards clinical BHL implementation, establishing the endothelialization of a whole assembled oxygenator consisting of multi-layer HFMs framed by a housing. For this approach, our established, standardized in vitro endothelialization protocol for single-layer HFMs might not be transferrable, as assembling a multi-layer oxygenator is associated with potentially cytotoxic or rather EML-damaging manufacturing steps, e.g., EML exposure to ambient air or mechanical stress during the potting procedure [27]. Therefore, we elaborate an alternative protocol for EML establishment after oxygenator assembly is completed and all blood-contacting surfaces are secured against the cell-detrimental environment. For this, we use a model oxygenator (MOx) that contains the relevant features of ECMO oxygenators, such as PMP HFM and polysulfone housing, and systematically identify the influence of specific variables, i.e., cell seeding concentration, rotational axis during seeding procedure, seeding time and culture medium conditions, which affect EC viability and adhesion and also the formation efficiency of the EML.

## 2. Materials and Methods

### 2.1. Endothelial Cell Culture and HFM Sample Preparation

#### 2.1.1. Isolation, Cultivation and Characterization of hCBECs

EC harvesting from human cord blood (hCBECs), cultivation and characterization were performed as described previously [28]. Briefly, after receiving informed consent, heparinized human cord blood was collected and diluted 1:1 with Dulbecco’s phosphate-buffered saline (DPBS) for density gradient centrifugation using Biocoll solution (Bio & Sell GmbH, Feucht, Germany). Afterwards, mononuclear cells (MNCs) were collected and seeded on gelatin-coated culture dishes and cultivated in endothelial growth medium 2 (EGM-2) (Lonza, Cologne, Germany) supplemented with 20% fetal bovine serum (FBS) under standard culture conditions (37 °C, 21% O_2_ and 5% CO_2_) until endothelial colony-forming cells were visible. Thereafter, the medium was changed to EGM-2 supplemented with 2% FBS. After reaching 80–90% confluence, hCBECs were enzymatically detached using 0.5%/0.2% trypsin/ethylenediamine tetra-acetic acid (EDTA) (Bio & Sell, Feucht, Germany), counted by the CASY TT cell counter (OLS, Bremen, Germany), subjected to assays for cell characterization and re-seeded at a cell density of 8 × 10^3^/cm^2^ for ongoing cell culture. hCBEC characterization was carried out by detection of the endothelial cell surface marker CD31 via flow cytometry and by expression analysis of endothelial-specific genes via qRT-PCR, as described previously in more detail [28]. After confirmative characterization, hCBECs were designated as ECs throughout this manuscript and used for the following experiments.

#### 2.1.2. HFM Fibronectin Coating for Reliable Endothelialization

Applying our previously established protocol [22], non-coated PMP HFMs (OXYPLUS™ 90/200, 3M/Membrana, Wuppertal, Germany) were sterilized with ethylene oxide (ethylene oxide sterilization) and trimmed in smaller patches for the following fibronectin (FN) coating process. Therefore, HFM patches were immersed overnight at 4 °C in human FN solution (lyophilized fibronectin, AdvancedBiomatrix, CA, USA) to enable complete coverage of the otherwise hydrophobic, cell-repellent PMP surface, securing a viable and confluent EML on the HFMs [22].

### 2.2. Application of Standardized In Vitro Protocol for HFM Endothelialization for Model Oxygenator Assembly

#### 2.2.1. Standardized In Vitro Endothelialization Protocol for Single-Layer HFMs Suitable for MOx Size

As previously described, we applied our well-established in vitro HFM endothelialization protocol, which was also optimized for our MOx, designed and constructed to simulate clinically applied ECMO oxygenators [29] (MHH Forschungswerkstatt, Hannover Medical School, Germany). To better understand MOx construction, a schematic exploded view (Figure 1E) and photography were included (Figure 1F). Summarizing, appropriately tailored FN-coated HFM patches were mounted in polycarbonate anchoring frames and introduced into 50 mL syringes (B. Braun, Melsungen, Germany) filled with 25 mL EC suspension containing 3 × 10^6^ ECs. To ensure optimal cell distribution, HFMs were set under 1 rpm rotation around the axis in the longitudinal direction of the fibers for 4 h using a roller device (Greiner, Frickenhausen, Germany) [20]. Afterwards, HFMs were taken from the syringes and incubated under static culture conditions for an additional 48 h in cell culture dishes (Greiner Bio-One, Frickenhausen, Germany) filled with EGM-2. Thereafter, HFMs were ready for assembly into the MOx.

#### 2.2.2. Transfer of In Vitro Reendothelialized HFMs for MOx Assembly

In general, for our MOx assembly, one in vitro endothelialized HFM patch was sandwiched between multiple silicone and steel frames under ambient air exposure (Figure 1E) until the MOx was closed and gently filled with culture medium. In accordance with our experiences, endothelialized HFM patches were exposed to ambient air for 1, 5 and 10 min to assess the effect of air exposure on EML viability caused by the time needed for single- or rather multi-layer MOx assembly. Afterwards, HFM patches were directly placed into new dishes for fluorescence microscopical assessment of viable ECs using 1 µM calcein AM (Sigma-Aldrich, Taufkirchen, Germany) and 10 µg/mL Hoechst 33342 (Sigma-Aldrich, Taufkirchen, Germany) for 30 min at 37 °C in EGM-2. Images were acquired with a ZEISS SteREO Discovery V8 microscope (ZEISS, Jena, Germany) equipped with a camera (AxioCam IcM1, Zeiss, Jena, Germany).

### 2.3. Establishment of the Optimal Endothelialization Protocol for Assembled MOx, Consisting of HFMs and Housing

#### 2.3.1. Identifying the Optimal EC Concentration for Complete MOx Endothelialization

To identify the optimal number of ECs per applied culture medium, two relevant core data were obtained. First, the total surface area of the MOx, consisting of HFMs and housing, resulted in 19 cm^2^ to be endothelialized, and, second, the total MOx volume was 4 mL. Based on our above-mentioned standard protocol for FN-coated HFM endothelialization, facilitating a confluent EML [22], we used an EC density of 1.9 ± 0.81 × 10^4^ ECs/cm^2^ as reference, which was consecutively named as the one-fold seeding concentration (1 × SC). Following that, we assessed EC coverage using sequentially redoubled cell concentrations, i.e., two-fold (2 × SC; 3.8 × 10^4^ ECs/cm^2^), four-fold (4 × SC; 7.6 × 10^4^ ECs/cm^2^) and eight-fold seeding concentration (8 × SC; 15.2 × 10^4^ ECs/cm^2^), respectively. By de-airing the MOx, each EC suspension was slowly injected through the access port on the inlet side (Figure 1E,F). Analogous to our above-mentioned, optimized protocol (see Section 2.2), the MOx was positioned inside an acrylic glass cylinder and fixed with silicone stoppers, enabling 1 rpm rotation around the axis in the longitudinal direction of the fibers for 6 h at 37 °C, subsequently referred to as rotational axis 1 (RA1). After the seeding procedure, the culture medium was removed from the MOx, and the respective number of non-attached cells of each cell concentration was determined by the CASY TT cell counter to indirectly calculate the number of adherent ECs. In order to directly verify the number of adhered ECs after 6 h using 8 × SC, the MOx was disassembled, and ECs were detached enzymatically from the HFMs and housing using trypsin/EDTA solution followed by centrifugation at 300× g for 3 min for cell counting using the CASY TT cell counter. To assess the distribution of adhered ECs on the different available surfaces after incubation at different cell concentrations, the MOx was disassembled, and the HFMs and housing were stained separately using calcein/Hoechst 33342 and quantified by fluorescence microscopy (see Section 2.2.2).

#### 2.3.2. Identifying the Optimal MOx Rotational Axis for Optimal Seeding Procedure

In comparison to RA1 (see Section 2.3.1), where the MOx revolved around the longitudinal direction of the fibers, we analyzed the impact on EC distribution in the MOx using rotational axis 2 (RA2), which was transversal to RA1. For this, in each case, only the optimal concentration of 8 × SC was used for the 6 h rotational seeding procedure, followed by qualitative fluorescence analysis of EC distribution on HFMs and housing (see Section 2.2.2). The number of adherent ECs was calculated from cells that remained in suspension, as described in Section 2.3.1. Direct cell counts of ECs on HFMs and housing were determined (see Section 2.3.1) and are included in the results described in 3.2.3. for comparison. To additionally quantify the successfully endothelialized areas in relation to the total available surface area, three random regions of interest (ROIs) were selected from calcein-stained images of both HFM sides (i.e., inlet- and outlet-facing) and independent triplicates using the image-processing program ImageJ (National Institute of Health, Bethesda, MD, USA), in particular, the integrated surface area measurement tool. Data were shown in combination with results obtained in Section 3.2.4. 

#### 2.3.3. Identifying the Ideal Rotation Time for Optimal Seeding Procedure

Here, the previously used 6 h rotation time with 8 × SC and RA1 and RA2 was compared to EML distribution on HFMs and housing after the same seeding procedures with 24 h rotation time (see Section 2.3.2). Additionally, respective MOxs were disassembled for direct cell count of adherent ECs after their enzymatic detachment (see Section 2.3.1).

#### 2.3.4. Identifying the Optimal Cell Culture Conditions during Seeding Procedure

##### Analysis of Culture Medium Conditions within Different Seeding Procedures

Due to various grades of metabolic cell activity within the different seeding procedures (8 × SC, RA1 vs. RA2, 6 h vs. 24 h rotation time), the constitution of the respective cell culture media was analyzed for their pH and concentrations of glucose and lactate, and also pO_2_ and pCO_2,_ using 100 µL samples in the blood gas analyzer (BGA) ABL90 FLEX (Radiometer, Brønshøj, Denmark).

##### Impact Analysis of Cell Culture Medium Exchange during Seeding Procedure

Assuming that physiologic cell culture conditions may support sufficient MOx endothelialization, a medium exchange protocol was also analyzed. Here, as part of the seeding procedure, using 8 × SC with RA2 for 24 h, complete cell medium exchange (MExch) and associated BGA analysis were performed every 6 h.For this, fresh medium was slowly injected through the inlet port and, therefore, flushed the consumed medium through the opposed outlet port. To further analyze the impact of MExch, the apoptotic levels of ECs on HFMs or the housing were determined by an annexin V/propidium iodide (PI) apoptosis assay (Miltenyi, Bergisch Gladbach, Germany) using a flow cytometer (MACSQuant 10, Miltenyi, Bergisch Gladbach, Germany). Briefly, ECs were, therefore, separately detached from HFMs or housing (see Section 2.3.3), and 1.5 × 10^4^ events of the main population were selected for analysis. Non-stained cells cultured on tissue culture plastic (TCP) were used as negative controls and determinants of the threshold of annexin-V- and PI-positive cells.

In order to assess EML formation and to quantify successfully endothelialized areas after MExch in comparison to after seeding procedures without MExch, HFM patches and housing were analyzed by fluorescence microscopy and quantitative analysis, as previously described (see Section 2.2.2 and Section 2.3.2). Additionally, endothelial-specific junction protein VE-cadherin was stained via immunofluorescence detection in order to confirm the intercellular connection of cells within the confluent EML. Therefore, HFMs were removed from the MOx directly after the respective seeding protocol and fixed in 4% paraformaldehyde (PFA)/DPBS for 10 min at room temperature (RT). Using a biopsy punch cutter, round samples (∅: 0.8 cm) were cut from the HFMs and transferred into vessels with DPBS. ECs residing on the HFMs were permeabilized and blocked for nonspecific antibody binding by incubation with 0.25% Triton-X100 diluted in Tris-buffered saline and supplemented with 5% donkey serum for 20 min at RT. Then, the samples were rinsed three times with DPBS and incubated with the primary anti-VE-cadherin antibody (diluted at 1:50 in 1% bovine serum albumin (BSA) in DPBS *w*/*o* Ca^2+^/Mg^2+^; AbD Serotec, Puchheim, Germany) for 1 h at RT. After three washing steps, fluorescence-labeled secondary antibodies, goat anti-mouse Cy2 (JacksonImmuno Research, Cambridge, UK), were applied for 1 h at RT in the dark. Nuclei were stained with Hoechst 33342 dye (10 μg/mL) added to the secondary antibody incubation for the last 20 min. ECs on FN-coated glass slides were used in parallel as negative controls, applying isotype-matching antibodies. After the staining procedure, samples were transferred into glass-bottom dishes (MatTek, Ashland, MA, USA) and assessed under a confocal laser scanning microscope (CLSM SP8-system, Leica Microsystems, Wetzlar, Germany). At the ROIs, z-stacks of images were generated and visualized as a volume projection image using ImageJ (Z-project, maximum brightness). For the visualization of the VE-cadherin signal, magenta was selected as a false color; nuclei were shown in blue.

### 2.4. Statistical Analysis

Statistical analysis was performed using GraphPad Prism Version 9 for Windows (GraphPad Software, San Diego, CA, USA). For the comparison of multiple groups, an ordinary one-way ANOVA test was applied. Statistical significances between two groups were calculated using unpaired, two-tailed *t*-test. Values are shown as means and standard deviation (SD). Differences were considered significant at *p* < 0.05. For each analysis, datasets from three independent replicates (*n* = 3) were used. Statistical significances are indicated as asterisks (* *p* < 0.05, ** *p* < 0.01, *** *p* < 0.001, **** *p* < 0.0001).

## 3. Results

### 3.1. Air Contact during Transfer of In Vitro Endothelialized HFMs for MOx Assembly Resulted in Significant Cell Damage

According to the fluorescence microscopy images, the surface of the HFMs was covered completely with a viable and confluent EML when our standardized in vitro endothelialization protocol was applied (Figure 1A). However, when the transfer step from the culture dish into the MOx was simulated by exposure of the seeded HFMs to ambient air for 1 min (Figure 1B), EC viability was predominantly affected in the central area of each individual fiber, as indicated by the loss of calcein fluorescence. However, the signal for the DNA-intercalating dye Hoechst 33342 was still detectable in these areas, confirming that ECs were not washed or worn away by effects other than cell death. Depending on the prolongation of the air contact to 5 (Figure 1C) and 10 min (Figure 1D), pronounced detrimental effects with more dead ECs in the central areas were observed.

### 3.2. Succesful Protocol Establishment for Complete Endothelialization inside the Fully Assembled MOx

#### 3.2.1. Eight-Fold Cell Concentration Resulted in Best MOx Endothelialization

To identify the optimal EC concentration for efficient MOx, consisting of HFMs and housing, endothelialization suspensions with sequentially increasing EC concentrations, i.e., 1 × SC−8 × SC, were applied and analyzed. Fluorescence microscopical assessment of ECs on HFMs and housing 6 h after the seeding procedure using RA1 with 1 × SC revealed viable and adherent ECs, which, however, were not dispersed uniformly all over the surface areas (Figure 2A–D). Instead, ECs were predominantly located on the areas between the fibers where the surfaces of the neighboring fibers directly faced each other. In contrast to the samples assessed after ambient air contact (Figure 1B–D), stained cell nuclei in areas without viable ECs were not observed. On both inlet- (Figure 2A) and outlet-facing sides (Figure 2B) of the HFMs, similar EC distribution patterns were noted. EC distributions on both sides of the housing were remarkably different, with only a few and singular adherent ECs on the inlet-facing part of the housing (Figure 2C) and a higher cell density with multiple colony-like ECs on the outlet-facing part (Figure 2D).

By sequentially doubling EC concentrations, i.e., 2 × SC, 4 × SC and 8 × SC, respectively, fluorescence microscopy revealed correlating, increased numbers of adherent ECs on HFMs in general (Figure 2E,F; Appendix A). However, it was observed that the concentration of adherent ECs on the inlet-facing HFM sides (Figure 2E; Appendix A) was always higher compared to that on the outlet-facing ones (Figure 2F; Appendix A). Applying 8 × SC resulted in a close-to-confluent monolayer on the inlet-facing HFMs (Figure 2E), with about 80% cellular coverage (Figure 3G). When assessing EC distribution on the housing sides, it was noted that, regardless of the applied EC concentrations, the frequency of adherent ECs on the inlet-facing side of the housing was very low (Figure 2G; Appendix A). In contrast, the number of adherent ECs found on the opposed outlet-facing side seemed to increase with higher EC concentrations (Figure 2H; Appendix A). At 2 × SC, instead of a confluent monolayer, ECs formed clusters of aggregated cells on the outlet-facing housing side (Appendix A). This observation was more prominent when 4 × SC (Appendix A) and 8 × SC cell suspensions (Figure 2H) were applied. The formation of a (sub-)confluent EML, not even on a few areas over the housing sides, was not detected, regardless of the injected cell concentrations. Thus, quantification of confluent areas on the housing sides was omitted for that reason.

When quantifying the number of viable ECs remaining in the cell suspension after 6 h, it was noted that the total number of adherent ECs also increased with the applied EC concentration. According to this, regardless of the applied EC concentration, about 10–11% of the initially injected ECs did not adhere either to HFMs or to the housing (Figure 2I). In order to figure out onto which of the two surfaces ECs were predominantly adhering to within 6 h and 8 × SC, ECs were detached either from the housing (for technical limitations from both sides together) or the HFMs and counted directly. Interestingly, 1.41 × 10^5^ ± 0.68 ×10^5^ ECs were retrieved after enzymatical detachment from the HFMs, while 7.71 × 10^5^ ± 1.93 × 10^5^ ECs resided on the housing (Figure 4I).

#### 3.2.2. Transversal Rotation Axis Improved EC Distribution in MOx

Considering the inhomogeneous EC distribution on the opposing housing and HFM sides, we hypothesized that the EC suspension needed to be dispersed more efficiently within the MOx and, therefore, changed the rotational axis for seeding procedure from longitudinal to transversal on the roller mixer. Indeed, applying RA2 for 6 h resulted in a homogenous distribution of viable and adherent ECs on the HFMs (Figure 5A,B), proven by the calcein/Hoechst 33342 staining. In particular, the outlet-facing HFM side was covered with a near-to-confluent EML under RA2 (Figure 5B) compared to samples endothelialized under RA1 (Figure 2E,F). To express this in numbers, only 49.87 ± 10.29% of the available outlet-facing HFM surface area was covered with ECs after RA1, whereas changing the rotation axis to RA2 resulted in coverage of 84 ± 2.48% (Figure 3G). Additionally, the total number of ECs that could be detached from the HFMs increased from 1.41 × 10^5^ ± 0.68 × 10^5^ ECs to 2.15 × 10^5^ ± 0.46 × 10^5^ ECs (Figure 4I). However, the apparent seeding efficiency of the housing indicated an improved EC attachment, especially on the inlet-facing side (Figure 5C), but it was still not optimal. ECs were still aggregated in clusters and spheroids, indicating incomplete cell adhesion to the housing surface (Figure 5C,D). Nevertheless, quantification of total ECs detached from the housing sides was not significantly increased (RA1: 7.71 × 10^5^ ± 1.93 × 10^5^ ECs vs. RA2: 6.19 ± 1.62 × 10^5^ ECs; Figure 4I). Summarizing, although fluorescence microscopy presented improved MOx endothelialization, also supported by the decreased number of nonadherent ECs, from 3.14 × 10^5^ ± 5.6 × 10^4^ using RA1 to 1.67 × 10^5^ ± 5.16 × 10^4^ (6% of 8 × SC) using RA2, this reduction of nonadherent ECs after changing the rotation axis compared to the former setting was statistically not significant (*p* > 0.05) (Figure 5G).

#### 3.2.3. Prolonged Rotational Seeding under RA2 Enhanced EC Attachment in the MOx

When extending the seeding procedure from 6 h to 24 h with 8 × SC and RA1, calcein/Hoechst 33342 staining of ECs revealed similar results regarding the distribution of viable cells on the MOx’s surfaces as those found after 6 h (Figure 2E–H and Figure 4A,B).

On the inlet-facing HFMs, 76.47 ± 13.64% of the available surface area was endothelialized after 24 h compared to 84.80 ± 2.97% after 6 h, while the measured confluent HFM areas of the fiber surface on the opposing HFM patch side slightly increased from 49.87 ± 8.40% after 6 h to the still non-optimal 65.57% ± 5.75% after 24 h (Figure 3G). The increased seeding time resulted in improved EC adhesion on the outlet-facing surface of the housing, where ECs showed a flattened morphology and started to form a confluent EML (Figure 4D). Again, however, only a few to no adherent ECs were detected on the inlet housing side (Figure 4C) and did not allow for reliable quantification of confluent areas. Quantification of the total EC numbers detached from the HFMs and both housing sides separately after 6 h and 24 h also showed no statistically significant difference, although the average total number of ECs detached from the housing after 24 h trended towards higher cell counts compared to 6 h of rotational seeding using RA1 (Figure 4I).

When assessing the distribution of viable ECs after 24 h of applying RA2, a remarkable improvement was detected. While EML formation on the inlet-facing HFMs remained optimal, now, the outlet-facing HFMs and both housing sides were also covered with a near-to-confluent EML (Figure 4E–H). Expressed in numbers, the endothelialized HFM area facing the outlet increased significantly from 86.72 ± 2.03% after 6 h to 95.61 ± 1.44% after 24 h (Figure 3G). However, interestingly, no significant increase in the total number of ECs was recorded when detached from the housing sides (RA1: 9.58 × 10^5^ ± 2.49 × 10^5^ vs. RA2: 7.04 × 10^5^ ± 2.97 × 10^5^ ECs; Figure 4I).

#### 3.2.4. Physiological Cell Medium Conditions were Required for Sufficient MOx Endothelialization

##### Prolonged Seeding Procedure was Associated with Significant Medium Consumption, Necessitating Periodic Medium Exchange within 24 h

Using 8 × SC with RA1 vs. RA2 and 6 h vs. 24 h of seeding procedure, each constellation indicated the expected cell metabolic-associated medium consumption but with different pronounced levels of all measured concentrations (Table 1 and Table 2). Comparison of the 6 h seeding procedure using RA1 rather than RA2 revealed a pH drop from 7.40 to 7.13 vs. 7.43 to 7.02 and a comparable glucose consumption of 1mmol/L within 6 h but higher lactate production in RA2 up to 3.83 mmol/L (Table 1). Oxygen and carbon dioxide changes were not notable for this setting (Table 2). By extending the seeding procedure time to 24 h, pH further decreased to 6.78 using RA1 and to 6.71 using RA2. The respective pH drops correlated with a more pronounced lactate level increase and glucose consumption of up to 60% for all 24 h setups. Nevertheless, longer incubation inside the closed MOx resulted in a lower oxygen concentration in the culture medium, where, after 6 h under RA1 and RA2, oxygen was reduced by 21% and 24.5%, respectively, while, after 24 h, the reduction of oxygen was 27% and 29.3% of the initially available oxygen at T_0_. Regarding carbon dioxide, a time-dependent increase was also observed, but it never reached hypercapnic conditions (Table 2).

By periodic medium exchange every 6 h during the 24 h seeding procedure with RA2, glucose and oxygen levels were replenished, while the accumulated lactate and carbon dioxide levels were depleted. All measured levels after 24 h were comparable to the levels after the 6 h seeding procedure, for example, pH after 24 h was less acidic at 6.94 in the medium exchange (MExch) setup and comparable to the pH after 6 h at 7.02 compared to 6.71 without MExch.

##### Medium Exchange during 24 h Seeding Procedure Alleviated Apoptosis

To assess apoptosis levels, EC populations were separately enzymatically detached from HFMs or the housing after the 24 h seeding procedure using RA2 and subjected to a flow cytometric annexin V/PI assay (Figure 6). In order to possibly identify the consumed medium as a source for apoptosis induction, samples from the MExch group were also analyzed. When compared to ECs cultured under standard conditions on TCP in parallel, the fraction of viable ECs detached from HFMs without MExch was significantly reduced (TCP: 93.95 ± 1.22% vs. HFMs without MExch: 85.81 ± 2.82%, *p* < 0.05). In contrast, ECs of the MExch group showed no difference to the ECs cultured on TCP (HFMs with MExch: 93.74 ± 2.52%). The fraction of viable cells among ECs residing on the housing was also improved when MExch was carried out (housing with MExch: 92.08 ± 4.20% vs. without MExch: 86.79 ± 2.28%), although these differences were not statistically significant. Moreover, MExch significantly alleviated the fraction of ECs on the HFMs entering apoptosis (HFMs without MExch: 10.73 ± 1.23% vs. with MExch: 5.00 ± 1.57%). Of note was that, under the same seeding conditions, no significant differences regarding viability or apoptosis between ECs residing on the HFMs or housing were found.

##### Medium Substitution during 24 h Seeding Procedure Did Not Affect EML Integrity in Confluent Areas

Fluorescence microscopy of calcein/Hoechst 33342-stained ECs after the 24 h seeding procedure using 8 × SC, RA2 and medium exchanges demonstrated a nearly confluent EML on wide areas of both the inlet- and outlet-facing HFMs (Figure 3A,B), but corresponding quantification resulted in lower values for confluence with 56.35 ± 6.15% on the inlet-facing side compared to 71.77 ± 7.50% for the outlet-facing one (Figure 3G). However, the occasional lack of viable ECs in the center of some fibers seemed not to be caused by necrosis, as a fluorescence signal for nuclear dye was observed in neither of these areas (Figure 3A,B). In contrast to the HFMs, both housing surfaces were still well endothelialized (Figure 3C,D).

Nevertheless, confocal laser scanning microscopy of well-seeded, confluent areas of the HFMs for immunofluorescence detection of VE-cadherin did not reveal any notable differences in cell morphology or cell junctions within the HFM EML with or without MExch (Figure 3E,F).

In summary, by means of these experiments, an optimal protocol for establishing a viable and confluent EML on both HFMs and housing inside the assembled MOx was identified, defining the eight-fold cell concentration within the best seeding procedure, consisting of 24 h using RA2 and medium exchange, as optimal.

## 4. Discussion

Aiming towards BHL development on the basis of contemporary ECMO, both as an alternative to LTx and as a bridge to recovery or transplantation, general hemocompatibility of all blood-contacting surfaces is indispensable for securing reliable, long-term application. Within recent years, various studies have demonstrated the general feasibility of endothelializing the largest and main oxygenator component, the PMP HFMs [15,16,17,18,19,20,21,22,28], which are the most vulnerable to thromboembolic events [8]. For this, we established an optimized and standardized in vitro seeding protocol, including, inter alia, sufficient surface coating of the artificial PMP, enabling the formation of a viable, anti-thrombogenic and flow-resistant EML [22]. Then, focusing on the next step towards translational BHL application, we utilized our gained experiences from in vitro endothelialization outside the disassembled oxygenator to establish complete endothelialization of the ECMO oxygenator consisting of both HFMs and the surrounding housing.

However, the results showed that our current standard in vitro protocol seems not to be conferrable for that purpose, as it already failed on the required step of transferring the pre-seeded HFM from the culture dish to the oxygenator components for assembly, during which the ECs were exposed to ambient air. Without sufficient cell medium coverage, significant cell death was observed. This effect was more pronounced the longer the ambient air contact was. Accordingly, after 10 min of simulated air contact, the initial completely confluent and viable EML on the FN-coated HFMs almost died off completely, and this duration did not even reflect the needed time to assemble a multi-layer oxygenator consisting of around 70 HFMs (e.g., in the iLA [31] (Novalung/Xenios) or the Quadrox [32] (Getinge/Maquet)) for adequate patient support. Even 1 min of exposure, representing the minimum time for single-layer assembly by screwing all MOx elements together and thereby sealing the HFMs for gas exchange (Figure 1E), caused a significant endothelial viability decline. The absent signal for viable cells was probably caused by dehydration or air-induced cell death [33] and not by incorrect handling, as nuclei were detectable in areas without viable ECs, therefore, disproving that cell depletion was caused by mechanical wearing. Furthermore, cell damage was most pronounced in the central areas of each fiber, where the medium looked dried up, and, in contrast, inter-HFM spaces were still wetted with medium, and viable ECs were detectable. In addition to this air-exposure-induced EC damage, further detrimental conditions are expected within clinical oxygenator assembly, even during the most novel manufacturing processes [27] where fibers are positioned in the housing and subsequently filled with potentially cytotoxic casting compound for the potting procedure. This potting also includes the application of a centrifugal force, which is followed by a wash-out process, so both steps might be harmful to the EML on the HFMs. Finally, our standardized in vitro protocol only focused on HFM endothelialization but not on EML formation on the oxygenator housing, which, as the blood-contacting surface, also needs to obtain complete hemocompatibility. Hence, these limitations led to the necessity of developing a new endothelialization protocol that circumvents the drawbacks mentioned above, enabling EML formation on the whole oxygenator consisting of HFMs and housing. For this, it seemed appropriate to establish the endothelialization in an already assembled oxygenator.

To initially define the most important basic settings which have an impact on successful EML formation for this approach, we used our MOx, which was designed following clinically applied oxygenators and had already been used in former experiments to successfully simulate different oxygenator settings [22,29]. Additionally, the MOx material characteristics were comparable to those of oxygenators, as common PMP HFMs could be inserted, and the surrounding housing was made of a hydrophobic polymer, such as polysulfone [34,35]. Based on our many years of experience in the field of tissue engineering, we were already aware that successful EML formation on artificial surfaces depends not only on the appropriate surface coating, but also on the adequate EC concentration per available surface area and cell medium, the available time for cell adhesion, the proper cell distribution for homogeneous EC adherence and, finally, the maintenance of a physiologic environment during EML formation.

To first confirm that optimal EC adhesion on the HFMs was feasible, fibronectin coating was chosen, as previous experiments showed its superiority over other coatings, e.g., albumin/heparin [22]. Furthermore, we adapted the rotational seeding from 4 h [22] to 6 h to additionally enable sufficient endothelialization of the housing, as previously 4 h was shown to be crucial only for sufficient uniform cell distribution and adhesion of the HFMs. To obtain a first impression of the effectiveness of the parallel endothelialization of HFMs and housing inside the MOx, as reference, we also applied the one-fold cell concentration (1.9 ± 0.81 × 10^4^ ECs/cm^2^), representing the number of ECs/cm^2^, which, in accordance with our standardized protocol, established a completely confluent EML on the FN-coated HFMs, knowing well that the additional housing surface was not considered in this approach. Therefore, as expected, cell density and distribution across the surfaces were not optimal with regard to confluent EML formation, but the results fortunately confirmed viable EC adhesion on both HFMs and housing. By successively doubling the applied cell concentration by up to eight-fold (15.2 × 10^4^ ECs/cm^2^), respectively, endothelialization increased visually, while the calculated number of adhered cells also increased accordingly. However, using this indirect determination of adhered cells did not match with the cell counting after enzymatic detachment from the MOx surfaces. This was likely due to the fact that the number of single ECs in the suspension after 6 h was underestimated, since cell aggregates may have been formed that were not measured by the automated cell counter. However, we accepted this as a systemic error, as this effect was true for all experiments at different seeding concentrations within 6 h. With respect to the cell distribution, the inlet-facing HFM surfaces demonstrated a viable and nearly confluent monolayer, while the outlet-facing HFMs and the juxtaposed housing of the inlet port were not adequately endothelialized, demanding further protocol adaption.

Hypothesizing that the observed non-uniform EC distribution resulted from suboptimal EC suspension mixing inside the MOx during longitudinal rotation, we also analyzed the impact of the transversal rotation axis, perpendicular to the parallel-aligned HFM fibers, first for the 6 h seeding procedure (Figure 5E,F). Overall, a viable and confluent monolayer could be observed on both HFM sides, and the area covered with a confluent EML almost doubled on the outlet-facing side, possibly due to the fact that transversal rotation supported the endothelial perfusion through the inter-HFM spaces compared to longitudinal rotation with perfusion only along or rather in parallel to the HFMs. For the endothelialization of the housing, the rotational axis seems to play only a minor role, with a tendency to prefer the longitudinal rotational axis. Although a few more viable and adherent ECs were detected on the housing, especially on the inlet-facing side, a large EC fraction did not spread over the surface and, instead, remained in a spheroidal shape; they aggregated or rather indicated incomplete adherence, making direct and reliable quantification of cellular confluence impossible. One reason for this difference in cellular behavior may be the different surface properties of polysulfone. While pre-coating with FN changed the initial hydrophobic and cell-repellent nature of the PMP to a cell-friendly surface, the non-coated polysulfone housing was possibly still more hydrophobic and did not allow EC adherence at the same speed as the FN-coated surface [36,37,38]. This hypothesis was also confirmed, as the extension of the seeding procedure to 24 h resulted in a more confluent EML on both housing sides. Therefore, in prospective experiments, we will analyze the impact of FN coating of the housing on speeding up efficient EML formation compared to FN-coated HFMs. If successful, the optimal endothelialization protocol of the oxygenator will include a FN pre-coating of the HFMs and housing, followed by the endothelialization. Interestingly, although the calcein staining also suggested a confluent monolayer on the HFMs after 24 h, the quantification of adherent ECs revealed fewer ECs compared to after 6 h, which may be explained by an altered cell adhesion pattern between the housing and the HFMs in favor to the housing. Summarizing the results so far, the most effective endothelialization of the entire MOx was achieved by the eight-fold cell concentration with the 24 h seeding procedure using the transversal rotational axis. Accordingly, when transferring the protocol to clinical oxygenators, rotation around all oxygenator axes (x, y and z) will presumably be needed to distribute the ECs evenly throughout the criss-cross, stacked, multiple HFM layers.

As sufficient nutrition supply during EML formation is mandatory, we analyzed the main relevant components during various seeding conditions. For both 6 h and 24 h procedures, adequate or rather even more than physiologically needed oxygen was available, while carbon dioxide was always below physiological reference values. This demonstrated that probably no additional gas exchange through the HFMs in parallel to EML formation is required. However, it attracted our attention that the extended 24 h seeding procedure under these hyperoxic conditions caused a significant accumulation of lactate with a significant, corresponding pH decrease, while glucose levels also declined to critically low levels [30]. However, a high lactate production rate from glycolysis under aerobic conditions is considered as a hallmark of endothelial proliferation and, therefore, a positive instead of a negative sign for cellular fitness [39,40], indicating physiological EC behavior. Nevertheless, on the one hand, this high metabolic activity can be faced by periodic medium exchange, and, on the other hand, it potentially slows down soon after achieving the completely confluent EML, which initiates contact inhibition of cell proliferation [41]. Although calcein staining after 24 h did not indicate massive cell death either on the HFMs or on the housing, the endothelial apoptosis rate on both surfaces was significantly elevated in comparison to the EC population cultivated in parallel as reference under culture conditions on TCPs. Furthermore, since apoptosis is an irreversible process [42], it was expected that a pronounced loss of viable cells would be visible when assessed only a little while after 24 h. As a consequence, we decided to introduce a medium exchange every 6 h, as lactate and glucose levels after 6 h were still acceptable for EC culture [43]. By this, we were able to maintain pH, glucose and lactate at less harmful levels, while, most importantly, the fraction of apoptotic ECs fell back to comparable levels to the reference. Nonetheless, quantification of adherent ECs, or rather fractions of endothelialized surface areas after three-fold medium exchange, indicated a slight loss of cells, possibly caused by wash out of ECs which did not adhere strong enough to the respective surface. To further optimize this protocol, different approaches are conceivable. On the one hand, a defined number of ECs could be added to the fresh medium to replace the lost ECs on the respective surfaces; on the other hand, a continuous medium perfusion with defined flow conditions could be applied. This would enable the continuous delivery of fresh ECs via the circulating cell medium, which would also secure stable culture conditions, supporting sufficient EML formation without apoptosis induction. In parallel, a slow medium perfusion could train the ECs towards flow conditions in advance by inducing the production of extracellular matrix proteins that support EC adhesion and flow resistance [44], which is one of the most important requirements for clinical application.

Summarizing, quantitative data analysis based only on independent triplicates, and, therefore, appearing as preliminary character, nevertheless emphasized the already clearly visible qualitative improvements in endothelialization of the assembled model oxygenator by the stepwise protocol adjustment very well and, thereby, underlined its validity, which enabled us to identify the most relevant factors for complete endothelialization. Even though the MOx does not reflect all the features of the clinically applied oxygenator becoming a BHL prototype, such as the size of the housing or rather the multiple HFM layer arrangement, the influencing factors identified in this study, i.e., EC seeding density, rotational seeding time and nutrient supply, are pioneering and equally important for achieving sufficient endothelialization of the prototype. First, the optimal EC seeding concentration of 8 × SC, i.e., 15.2 × 10^4^ ECs/cm^2^, allowed sufficient EML formation within 24 h on both the MOx HFMs and the housing. This result has high relevance when considering the endothelialization of commercially available ECMO oxygenators, such as the iLA (Novalung) [45] with an estimated surface area of about 1.35 m^2^, for which, accordingly, 2 × 10^9^ ECs would be needed. This fact dictates that our earlier described iPSC-differentiated or MHC-silenced ECs need to be considered for BHL seeding rather than autologous ECs directly isolated from the patient, which do not yield this needed number. Furthermore, viable and confluent EML formation depends on optimal EC distribution during rotational seeding, which can be performed around the x, y, z axis of commercially available oxygenators, in order to distribute the ECs throughout the criss-cross, stacked HFM multi-layers. Further, keeping the cell medium at physiological conditions, and, therefore, preventing apoptosis, is essential and can be achieved by both repetitive medium exchange or steady and slow perfusion, which may be even more beneficial, as it additionally induces some kind of “flow training”, resulting in extracellular matrix production [46] which is indispensable for a flow-resistant EML. By the successful translation of this protocol to commercially available and clinically applied oxygenators, the next important milestone towards clinical implementation of the BHL will be achieved.

## Figures and Tables

**Figure 1 bioengineering-10-00072-f001:**
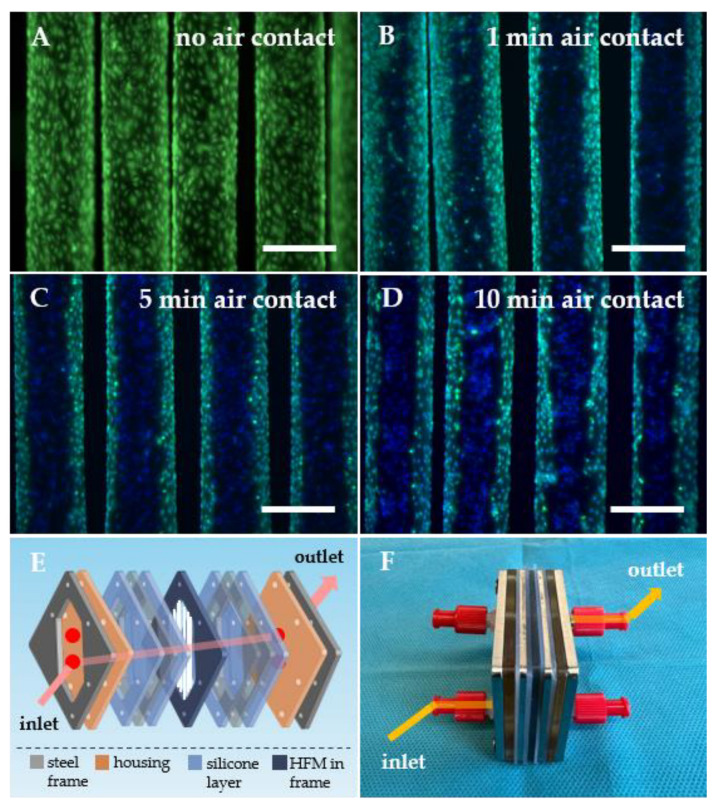
EC monolayer established on HFMs by standardized in vitro protocol was adversely affected by air contact during transfer for MOx assembly. Fluorescence microscopy for the detection of calcein-stained, viable ECs (green, A–D) covering the HFM surface without air contact (**A**) and after air exposure for 1 min (**B**), 5 min (**C**) or 10 min (**D**). Nuclei were counterstained with Hoechst 33342 (blue), scale: 400 µm. The exploded view illustrates the various layers of the model oxygenator (MOx) (**E**), supplemented by a photo (**F**); respective arrows demonstrate the direction of the cell suspension’s injection through the MOx.

**Figure 2 bioengineering-10-00072-f002:**
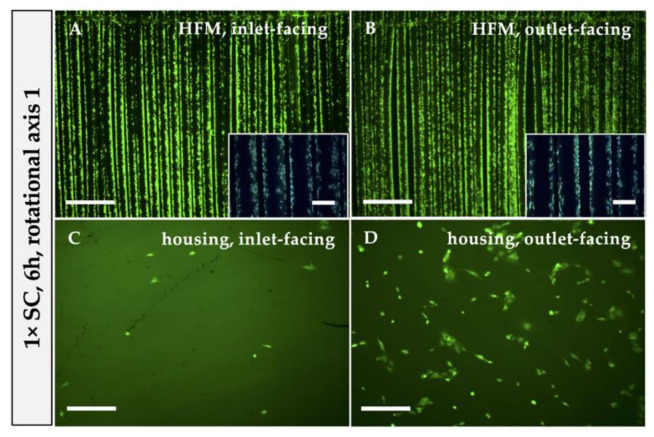
Improved seeding efficiency depended on applied EC concentration. Fluorescence microscopy was performed on HFMs (**A**,**B**,**E**,**F**) and the housing (**C**,**D**,**G**,**H**), facing the inlet (**A**,**C**,**E**,**G**) and the outlet (**B**,**D**,**F**,**H**), after seeding with 1 × SC (**A**–**D**) or 8 × SC (**E**–**H**) and using calcein (green)/Hoechst 33342 (blue) staining. Scale: 2 mm; insert boxes in A, B, E, F, scale: 400 µm. (**I**) Quantification of percent adherent ECs after incubation with different seeding concentrations. SC: seeding concentration.

**Figure 3 bioengineering-10-00072-f003:**
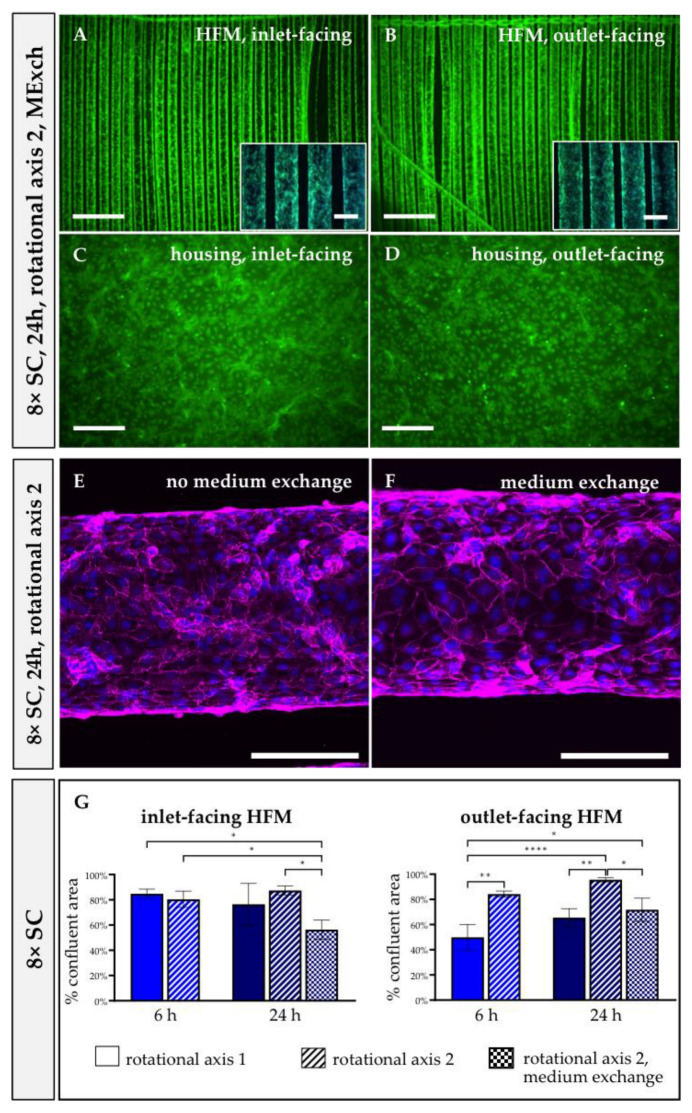
Confluent EML on HFMs remained unaffected after medium exchange during 24 h rotational seeding. Following seeding procedure using 8 × SC for 24 h and RA2, calcein (green) and Hoechst 33342 (blue) staining was performed, and fluorescence microscopy images were taken from HFMs (**A**,**B**) and housing (**C**,**D**) either facing the inlet (**A**,**C**) or outlet (**B**,**D**). Scale: 2 mm, insert boxes in A, B, scale: 400 µm. (**E**,**F**) CLSM images for the detection of VE-cadherin (magenta) on HFM samples cultivated without (**E**) or with MExch (**F**). Nuclei were counterstained with Hoechst 33342 (blue). Scale: 200 µm. (**G**) Quantification of the successfully endothelialized HFM area per total available HFM surface. SC: seeding concentration. * *p* < 0.05, ** *p* < 0.01, **** *p* < 0.0001.

**Figure 4 bioengineering-10-00072-f004:**
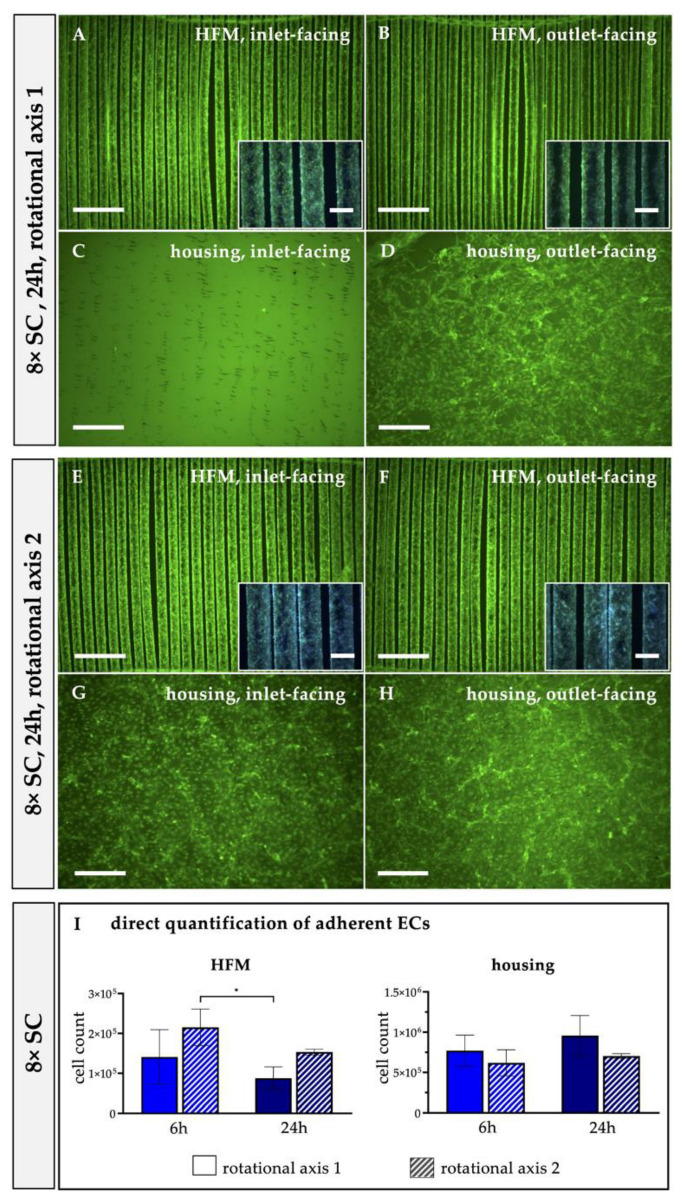
Twenty-four hour seeding procedure succeeded in complete MOx endothelialization when 8 × SC and RA2 were applied. Following staining with calcein (green) and Hoechst 33342 (blue), fluorescence microscopy images were taken from HFMs (**A**,**B**,**E**,**F**) and the housing (**C**,**D**,**G**,**H**), facing the inlet (**A**,**C**,**E**,**G**) and outlet (**B**,**D**,**F**,**H**), after seeding with 8 × SC for 24 h and applying RA1 (**A**–**D**) or RA2 (**E**–**H**). Scale: 2 mm, insert boxes in A, B, E, F, scale: 400 µm. (**I**) Direct quantification of adherent ECs on HFMs and housing after seeding procedure using 8 × SC for 6 h vs. 24 h with RA1 vs. RA2. SC: seeding concentration, RA: rotational axis. * *p* < 0.05.

**Figure 5 bioengineering-10-00072-f005:**
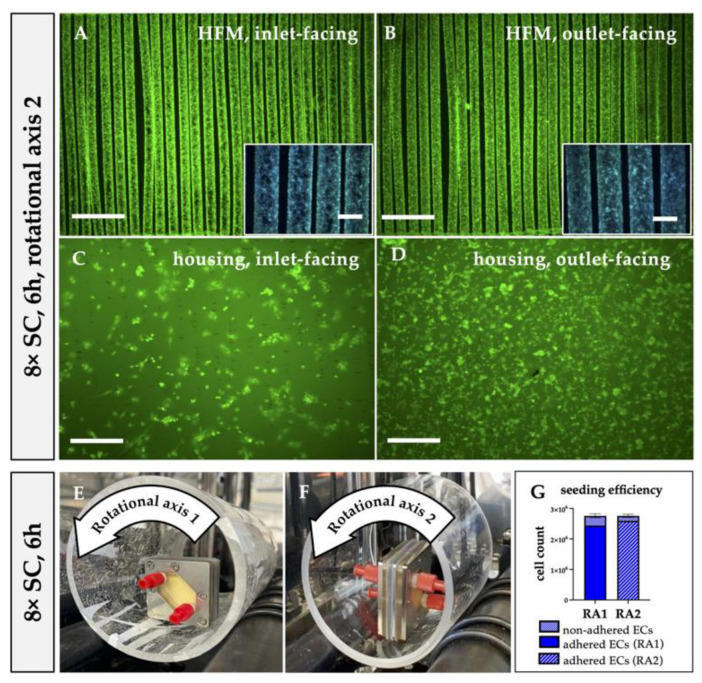
Transversal rotational axis for seeding procedure improved EC distribution. Following the injection of 8 × SC and changing from RA1 (**E**) to RA2 (**F**) for 6 h, fluorescence microscopy images were taken from HFMs (**A**,**B**) and housing (**C**,**D**) and facing inlet (**A**,**C**) and outlet (**B**,**D**) after staining with calcein (green) and Hoechst 33342 (blue). Scale: 2 mm, insert boxes in A, B, scale: 400 µm. Picture of the oxygenator position on the roller device during RA1 (**E**) and RA2 (**F**), respective silicone stoppers fixing the oxygenator in position were taken out for better visualization. (**G**) Quantification of percent adherent ECs of the complete MOx after incubation with 8 × SC for 6 h under RA1 and RA2. SC: seeding concentration, RA: rotational axis.

**Figure 6 bioengineering-10-00072-f006:**
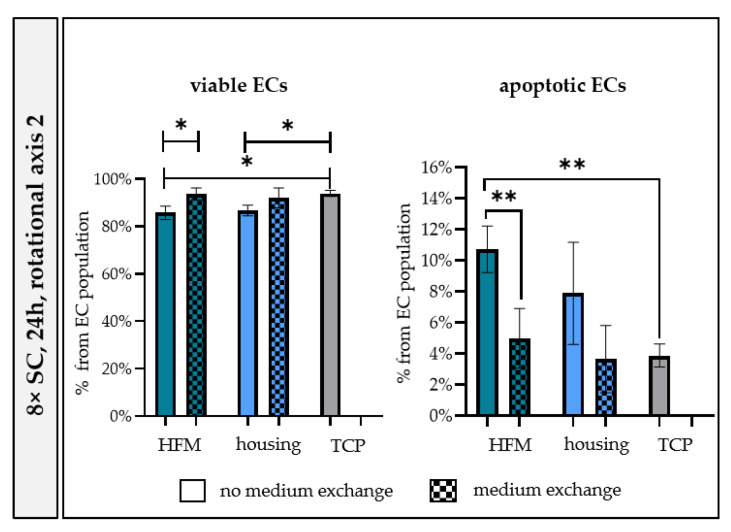
Medium exchange during 24 h seeding procedure recovered viability and mitigated apoptosis of ECs on MOx. Flow cytometry data of viable and apoptotic ECs obtained after rotational seeding around RA2 with 8 × SC for 24 h, with or without MExch. TCP: tissue culture plastics, SC: seeding concentration. * *p* < 0.05, ** *p* < 0.01.

**Table 1 bioengineering-10-00072-t001:** Analysis of pH, lactate and glucose in cell medium at different time points during different seeding conditions. Values are provided as means with standard deviation (SD) from independent triplicates. Reference values for venous blood, as prospectively perfused through the BHL, are in brackets [30]. RA1: rotational axis 1, RA2: rotational axis 2, MExch: medium exchange.

	pH (7.30–7.43)			Lactate (0.4–2.2 mmol/L)			Glucose (3.1–5.5 mmol/L)		
	Time [h]								
Seeding Condtionss	T_0_	T_6_	T_24_	T_0_	T_6_	T_24_	T_0_	T_6_	T_24_
6 h, RA1	7.40 ± 0.01	7.13 ± 0.02	_	0.57 ± 0.05	2.90 ± 0.08	_	5.37 ± 0.05	4.37 ± 0.05	_
6 h, RA2	7.43 ± 0.06	7.02 ± 0.02	_	0.50 ± 0.08	3.83 ± 0.05	_	5.30 ± 0.00	4.30 ± 0.08	_
24 h, RA1	7.36 ± 0.04	_	6.78 ± 0.02	0.50 ± 0.00	_	8.30 ± 0.14	5.30 ± 0.00	_	1.97 ± 0.05
24 h, RA2	7.35 ± 0.01	_	6.71 ± 0.02	0.40 ± 0.00	_	8.57 ± 0.71	5.17 ± 0.05	_	1.63 ± 0.26
24 h, RA2, MExch	7.33 ± 0.01	7.03 ± 0.01	6.94 ± 0.05	0.57 ± 0.05	3.53 ± 0.24	3.47 ± 0.48	5.10 ± 0.00	4.07 ± 0.09	4.17 ± 0.12

**Table 2 bioengineering-10-00072-t002:** Analysis of oxygen and carbon dioxide in cell medium at different time points during different seeding conditions. Values are provided as means with standard deviation (SD) from independent triplicates. Reference values for venous blood, as prospectively perfused through the BHL, are in brackets [30]. RA1: rotational axis 1, RA2: rotational axis 2, MExch: medium exchange.

	Oxygen (19–65 mmHg)			Carbon Dioxide (38–58 mmHg)		
	Time [h]					
Seeding Condtionss	T_0_	T_6_	T_24_	T_0_	T_6_	T_24_
6 h, RA1	173 ± 0.00	136.67 ± 3.40	_	17.77 ± 0.57	28.20 ± 1.27	_
6 h, RA2	195 ± 2.94	147.33 ± 1.25	_	17.50 ± 0.99	32.37 ± 1.56	_
24 h, RA1	171 ± 1.41	_	124.67 ± 9.46	19.47 ± 1.64	_	37.77 ± 1.87
24 h, RA2	175 ± 0.82	_	123.73 ± 20.98	18.83 ± 0.59	_	38.90 ± 2.76
24 h, RA2, MExch	177.67 ± 4.64	149 ± 5.35	140 ± 6.48	20 ± 0.71	32.80 ± 0.64	41.13 ± 3.73

## Data Availability

Not applicable.

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
