# Peer review of "Biohybrid lung Development: Towards Complete Endothelialization of an Assembled Extracorporeal Membrane Oxygenator"

_bioengineering, 2023, doi:10.3390/bioengineering10010072_

Round 1

Reviewer 1 Report

Very nice manuscript. The team is world leading in the work towards creating a hybrid lung. The authors should be commended for producing such a nice work. I have concerns or any corrections.

Reviewer 2 Report

The article is of major interest and provide new important technical progress. It is well presented and easy to read.

There is a point which should be improved, the statistics. It is not possible to make any statistical calculation with three values, the minimum is six and most of the tests required 12 to 21 values. To really reach the excellence the authors have to add more values or for some aspects limited the number to three indicating that is a preliminary aspect.

The values presented in table one are numerous and could be clearer. How many values are included in the mean and SD (n=  ). This could be improved in the table or the legend.

Reference 27: The chapter and the pages should be indicated.

Reviewer 3 Report

The manuscript reports about experiments of the authors to optimize the seeding conditions of endothelial cells (derived from human cord blood) on materials used for extracoprporeal membrane oxygenators such as hollow fiber membranes and the material of the housing. The main findings are an optimal cell density in the seeding suspension, the incubation time and a rotation protocol for an equal distribution of endothelial cells on these surfaces in an model oxygenator of 19 cm².

I regard the length of the manuscript as by far too long and suggest shortening of the introduction, the results and the discussion. Abbreviations should be carefully considered, since some are standard such as FN, EC, MNC etc, some are not. E.g. the abbreviation MO can be read as membrane oxygenator or model oxygenator. Please help the reader to read your paper in shorter time. In the results section I suggest to just briefly mention the results of Figue 1 (air contact) in one sentence in the results section. In Figure 2 I would skip the photos and maintain just part I. The photos in Figure 2, 3 and 4  can be combined showing e.g. the worst and the best result and describing the experimental differences in the text. Figure 4I can be included and briefly discussed in the discussion.  Table 1 should be omitted and described together with Figure 5 in the chapter on medium conditions.

The discussion should concentrate on the main findings in the manuscript in relation to the references cited. Own results previously published should be kept to a minimum. I miss the concession that the results presented here are far from the real application. This is the case a) for the small surface areas applied e.g. 19 m² membrane surface area in the model oxygenator as compred to > 1 m² in a clinical oxygenator, b) the important aspect of changing the hydrohobicity of the mebrane surface by fibronection and EC coating since a hydrophobic membrane surface is an absolute requirement for gas exchange by the membrane in blood c) no medium flow was used which will, depending on the flow conditions, alter EC adhesion and my result in shear stress related detachment of the cells. In addition, medium flow will enable to add nutrients and oxygen to the cell layers and control cell metabolism.

Also, I regard the time required for attachment as rather long since ECs adhere within 60 min to material surfaces. Finally, neither functionallity of the mebranes nor the EC layer was assessed in the manuscript such as proliferation, gene experesssion (by PCR) and improved hemocompatibility (the purpose of the whole project!).

Reviewer 4 Report

1. Introduction mention other the hollow fiber membranes (HFM) models of endothelization or angiogenesis-related.

2.Transversal rotational axis for seeding procedure- is microgravity a factor also? 

3. It would be interesting to see the impact of ROS in the Transversal rotational axis for seeding procedure- ECs seeding density.

4.VE-Cadherin staining in Figure 6 is critical to study outcomes. Mention the gap junction density chart also by doing an image analysis.

5. Mention its potential application in tissue engineering for example in below reference and cite it:

Ref.1. https://doi.org/10.1016/j.biomaterials.2020.119919
